# SHAPEGEN4D: TOWARDS HIGH QUALITY 4D SHAPE GENERATION FROM VIDEOS

**Jiraphon Yenphraphai**[*1,2]    **Ashkan Mirzaei**[1]    **Jianqi Chen**[3]    **Jiaxu Zou**[1]
**Sergey Tulyakov**[1]    **Raymond A. Yeh**[2]    **Peter Wonka**[1,3]    **Chaoyang Wang**[1]
[1]Snap    [2]Purdue University    [3]KAUST

https://shapegen4d.github.io/

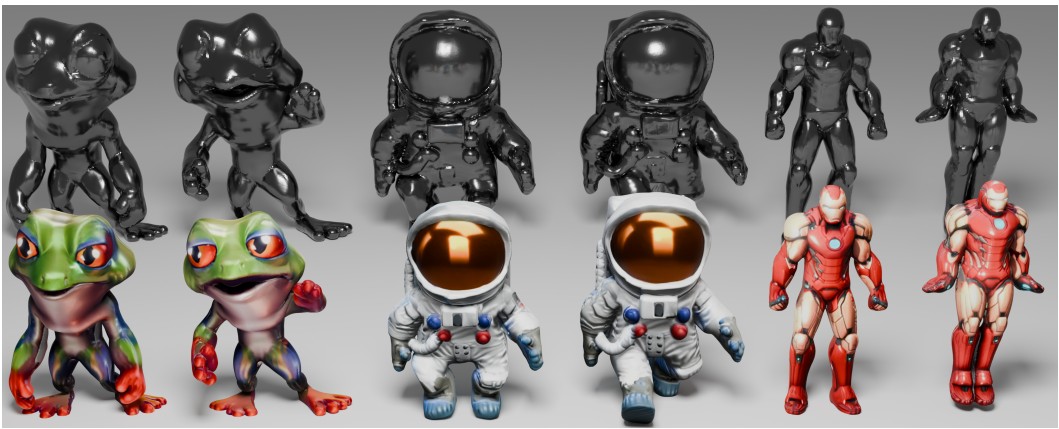

Figure 1: **ShapeGen4D** generates high-quality mesh sequences from input monocular videos.

## ABSTRACT

Video-conditioned 4D shape generation aims to recover time-varying 3D geometry and view-consistent appearance directly from an input video. In this work, we introduce a native video-to-4D shape generation framework that synthesizes a single dynamic 3D representation end-to-end from the video. Our framework introduces three key components based on large-scale pre-trained 3D models: **(i)** a temporal attention that conditions generation on all frames while producing a time-indexed dynamic representation; **(ii)** a time-aware point sampling and 4D latent anchoring that promote temporally consistent geometry and texture; and **(iii)** noise sharing across frames to enhance temporal stability. Our method accurately captures non-rigid motion, volume changes, and even topological transitions without per-frame optimization. Across diverse in-the-wild videos, our method improves robustness and perceptual fidelity and reduces failure modes compared with the baselines.

## 1 INTRODUCTION

Video-conditioned 4D shape generation aims to produce dynamic objects and their textures from an input video. With the rapid progress of generative models for images (BlackForestLabs, 2024), videos (Team, 2025), and 3D geometry (TencentHunyuan3DTeam, 2025a), this task has seen remarkable advances over the past two years. One line of work builds on score distillation sampling (SDS) (Poole et al., 2023) to optimize 4D shapes (Singer et al., 2023). However, SDS methods are fragile and computationally expensive. Subsequent approaches improve robustness and efficiency by adopting a two-stage pipeline (Zhang et al., 2024a; Xie et al., 2025; Wu et al., 2025a; Wang et al., 2025b;a): first generating multi-view videos with an image or video diffusion model, then reconstructing 3D or 4D geometry from these views. While faster and more stable than SDS, these

---

*Work done during an internship at Snap

methods remain limited by reconstruction errors and imperfections accumulated during multi-view generation, leading to suboptimal quality and efficiency.

More recently, inspired by the success of large-scale latent-diffusion transformers for 3D generation (TencentHunyuan3DTeam, 2025a; Xiang et al., 2025; Zhang et al., 2024b), two preliminary approaches have attempted to adapt pretrained 3D generative models for video-to-4D tasks. However, these methods either do not natively generate meshes in 4D or rely on optimization-based pipelines rather than fully feedforward generation. V2M4 (Chen et al., 2025) applies a 3D generative model independently to each video frame, followed by optimization to improve temporal smoothness and texture consistency. Despite these efforts, the method remains prone to artifacts in geometry, motion, and texture. In contrast, Gaussian Variational Field Diffusion (GVFD) (Zhang et al., 2025) generates the first frame with Trellis and trains a model to deform this initial geometry. This strategy has two key drawbacks: (a) geometry and texture are conditioned only on the first frame, ignoring new information revealed in later frames, and (b) the reliance on limited 4D training data,*e.g*., (Deitke et al., 2023), restricts the framework to rigid or near-isometric deformations, preventing it from handling topological changes or large volume variations such as growth or shrinkage.

In this work, we propose the first video-to-4D generation framework that directly produces dynamic 3D meshes. Building on recent advances in 3D generative models (TencentHunyuan3DTeam, 2025a; Xiang et al., 2025; Li et al., 2025a), our approach differs from concurrent work GVFD: rather than introducing a new modality such as deformation offsets of Gaussian particles, we generate a sequence of 3D meshes—a capability already learned by base 3D models—while explicitly addressing temporal consistency. This design accommodates variations in topology and relaxes constraints on the types of possible animations. More importantly, leveraging state-of-the-art pretrained 3D generators enables effective transfer of knowledge from large-scale 3D datasets, which are far more abundant than 4D datasets. With the generated mesh sequences, we can orthogonally apply registration and texturization to build animatable assets, *i.e*., a simpler and more stable process than generating animations from scratch, as in GVFD.

To realize this framework, we adapt 3D generative models to the 4D setting through **three key design choices**: (a) incorporating spatiotemporal attention in the shape diffusion transformer to capture temporal dependencies; (b) redesigning point sampling in the shape encoder to improve latent consistency and diffusibility; and (c) leveraging shared noise across frames to enhance temporal stability. We empirically verified that these design choices lead to high-quality 4D generation results; See teaser in Fig. 1.

**In summary, our main contributions are:**

- We propose the first video-to-4D generation framework that directly produces dynamic 3D meshes. Our method extends the architecture of a 3D shape generation method. The model is fine-tuned, rather than used as a black-box and extended with a separate network, like (GVFD) (Zhang et al., 2025), or combined with optimization, like V2M4 (Chen et al., 2025).

- To extend an SOTA 3D generation model to 4D generation, we propose several novel architectures and training innovations, including a spatio-temporal attention, redesigned surface point sampling, and noise sharing across frames.

## 2 RELATED WORK

**SDS-based 4D generation.** Score Distillation Sampling (SDS) (Poole et al., 2023) has emerged as a widely adopted approach for lifting 2D image diffusion models into 3D and 4D domains, with subsequent work advancing resolution, stability, and efficiency (Lin et al., 2023; Wang et al., 2023a;b; Tang et al., 2024b; Chen et al., 2023). A major research direction is the integration of heterogeneous diffusion priors (Qian et al., 2024; Shi et al., 2024), where text-to-image, multi-view, and video diffusion models are combined to promote both appearance fidelity and temporal coherence (Bahmani et al., 2024; Jiang et al., 2024b; Zhao et al., 2023; Jiang et al., 2024b). Other works (Zheng et al., 2024; Yuan et al., 2024; Ling et al., 2024; Li et al., 2024c; Yu et al., 2024) focus instead on enhancing the 4D representation itself, leveraging deformation fields, Gaussian splatting, and normal/depth priors to improve geometry quality and disentangle material properties. While SDS-based methods

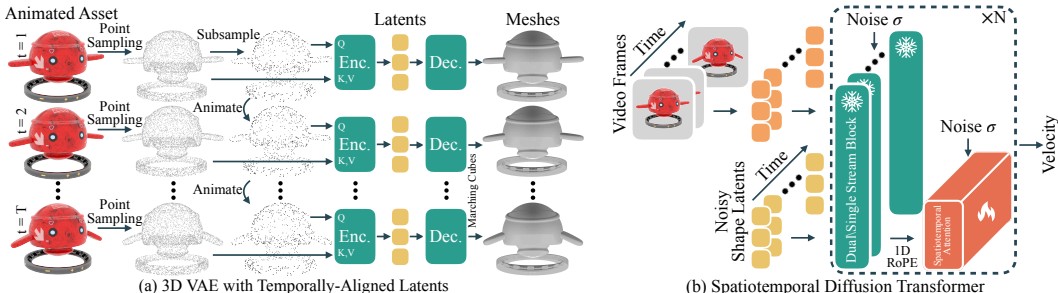

Figure 2: ShapeGen4D employs a flow-based latent diffusion transformer to generate a sequence of meshes from an input video. (a) A 3D VAE encodes shapes into latents by cross-attending subsampled query points with a dense point cloud. To encode a sequence of animated assets, query points are subsampled from the first-frame point cloud and then propagated through the animation to obtain query points for subsequent frames—yielding temporally-aligned latents. The decoder maps these latents to signed distance fields, which are then converted into meshes via marching cubes. (b) The spatiotemporal diffusion transformer interleaves frozen dual/single-stream transformer blocks from the base 3D generative model, which process hidden states for each frame independently, with learnable spatiotemporal attention layers that capture cross-frame dependencies and enforce temporal consistency in the denoised latents.

produce promising results, they remain computationally intensive and susceptible to artifacts such as Janus artifacts and oversaturation.

**Multi-View diffusion then reconstruction.** To bypass costly iterative optimization, recent 3D generation works (Li et al., 2024b; Zhuang et al., 2025; Xu et al., 2024; Shi et al., 2024; Huang et al., 2025) adopt a two-stage paradigm: first generating multi-view images, then reconstructing 3D representations with a feedforward model. Inspired by this, several recent 4D generation methods also follow a two-stage design. A key challenge lies in building multi-view video diffusion models that can generate frames across both time and viewpoints simultaneously. This is difficult due to limited training data. Early attempts (Zeng et al., 2024; Jiang et al., 2024b; Yang et al., 2025; Sun et al., 2025a) adapt multi-view image diffusion in a training-free manner to improve temporal consistency, but remain limited in robustness and the scale of motion they can handle. With advances in camera and motion control for video diffusion, works such as DimensionX (Sun et al., 2025b) and CAT4D (Wu et al., 2025a) show that multi-view videos can be generated via sophisticated procedures that alternate diffusion steps across time and view axes. A more fundamental solution is to train models that directly generate synchronized multi-view video grids. Research progress has moved from dynamic objects (Zhang et al., 2024a; Jiang et al., 2024; Li et al., 2024a; Xie et al., 2025; Yao et al., 2025) and humans (Shao et al., 2024) to general scenes, with notable works such as 4Real-Video (Wang et al., 2025b;a) and SynCamMaster (Bai et al., 2025). Despite these advances, the two-stage paradigm still suffers from accumulated errors due to inconsistencies in the generated video frames, making it difficult to reconstruct high-quality geometry.

**Direct 3D generation.** An alternative paradigm is to train generative models that directly produce 3D representations (Jun & Nichol, 2023). With the growing scale of 3D training datasets, diffusion-transformer–based direct 3D generators have emerged as the new state of the art. These models mainly differ in how they tokenize 3D shapes into diffusable latents. One line of work, following 3DShape2VectSet (Zhang et al., 2023), encodes unstructured point clouds into sets of unordered latent vectors. Large-scale foundation models such as Hunyuan3D 2.0 (TencentHunyuan3DTeam, 2025a;b), Step1X-3D (Li et al., 2025a), and others (Zhang et al., 2024b; Li et al., 2025b) build on this representation, achieving generalizable, high-quality geometry and texture with semantic- and UV-aware enhancements (Hunyuan3D, 2025). Another line of work leverages voxel grids and structured latents that decode into meshes, Gaussians, or radiance fields, as proposed by TRELLIS (Xiang et al., 2025) and extended by follow-ups (Wu et al., 2025b; Li et al., 2025c). Among these models, Step1X-3D (Li et al., 2025a) places particular emphasis on fidelity and openness, making it a suitable foundation for our work.

**Direct 4D generation.** While current direct 3D generators emphasize scalability, flexibility, and editability, extending them to direct 4D generation remains underexplored, largely due to the scarcity

of 4D data. L4GM (Ren et al., 2024), inspired by large multi-view Gaussian model (LGM) (Tang et al., 2024a), predicts multi-view images where each pixel represents a Gaussian particle conditioned on an input video. However, constrained by its non-diffusion backbone, small training set, and image-based representation, L4GM suffers from limited quality, generalizability, and relatively weak geometry. V2M4 (Chen et al., 2025) leverages per-frame meshes generated by state-of-the-art 3D models such as TRELLIS, but itself is not a direct method, relying instead on complex and fragile pipelines for mesh registration and geometry optimization. Concurrently, GVFD (Zhang et al., 2025) trains a diffusion model to generate deformation fields that drive Gaussian particles pre-generated by off-the-shelf 3D generators. Yet, due to limited 4D training data and restrictions on the possible deformations, learning deformation fields as a new modality yields weak generalization; in our experiments, GVFD generalized poorly. In contrast, our work deeply integrates with strong pretrained 3D generators to inherit their generalization ability, while leaving texturing to a simpler mesh registration step.

# 3 DYNAMIC MESH GENERATION

We present a flow-based latent diffusion model that generates mesh sequences capturing dynamic object motion, conditioned on a monocular video (see Fig. 2). Specifically, this involves extracting temporally-aligned latents by querying at the "same" surface location and introducing a spatiotemporal transformer for processing the sequence of frames.

## 3.1 TEMPORALLY-ALIGNED LATENTS FOR DYNAMIC SHAPE VAE

A compact, expressive, and well-regularized latent space is essential for training diffusion models (Skorokhodov et al., 2025). We build upon the VAE architecture adopted in recent 3D generative models, including Step1X-3D (Li et al., 2025a) and Hunyuan3D (TencentHunyuan3DTeam, 2025a), which encodes a sequence of meshes into a sequence of latent codes. The decoder then maps these latents into continuous geometric representations, *i.e.*, truncated signed distance fields. We first review the original VAE in **preliminary** and then describe our extension in **aligning latents**.

**Preliminary.** The VAE architecture we adopt follows the 3DShape2VectSet design (Zhang et al., 2023), which has been utilized in several state-of-the-art 3D generation models, including Step1X-3D (Li et al., 2025a) and Hunyuan3D (TencentHunyuan3DTeam, 2025a).

*Encoder:* Each mesh is first converted into a watertight form, and a dense set of points $\mathcal{P}$ is sampled from its surface. To create a fixed-length representation for the unstructured point cloud, the encoder cross-attends $\mathcal{P}$ with a set of query points $\mathcal{Q} = \text{FPS}(\mathcal{P})$, obtained via farthest point sampling (FPS). The resulting tokens are then processed by self-attention layers to produce latent codes. To better capture high-frequency details, $\mathcal{P}$ is augmented with salient points sampled along sharp edges, and each point is concatenated with surface normal features.

*Decoder:* The decoder first processes the latent codes through self-attention layers to extract shape features. For any 3D position $x$, the truncated signed distance value is predicted by cross-attending the positional embedding of $x$ with the decoded shape features.

**Aligning latents.** Naively encoding a mesh sequence $\{\mathcal{M}_1, ..., \mathcal{M}_T\}$ by encoding each frame independently produces temporally jittery latents. See Fig. 3 for illustration. As the sparse query set $\mathcal{Q}_t$ is sampled independently from $\mathcal{P}_t$ at each frame $t$, this leads to inconsistent query points across time. In our preliminary experiments, we found that such jittered latents make it harder for the diffusion model to learn smooth temporal dynamics.

To address this, we introduce temporal structure into the sequence of query sets $\{\mathcal{Q}_1, ..., \mathcal{Q}_T\}$. Instead of independent samples, we first sample $\mathcal{Q}_1$ from mesh $\mathcal{M}_1$, and then obtain subsequent sets by warping $\mathcal{Q}_1$ with the corresponding animation, *i.e.* $\mathcal{Q}_t = w_t(\mathcal{Q}_1)$, where $w_t$ denotes the deformation at the $t$-th frame. This ensures temporal alignment: each latent sequence corresponds to the same physical point on the deforming surface. Empirically, we find that aligned latents substantially reduce jitter and improve diffusion training.

In more detail, we sample query points directly from the original non-watertight mesh, where the animation is defined, rather than from the post-processed watertight mesh. Otherwise, establishing correspondences requires costly mesh registration.

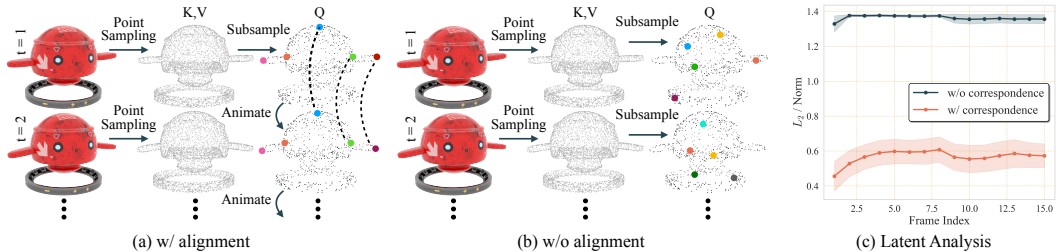

Figure 3: Illustration of latents with and without aligning query points across frames in (a) and (b). In (c), we visualize the average normalized $L_2$ difference between latents at the closest 3D positions across neighboring frames. We observe that with alignment, the $L_2$ difference is smaller, indicating that the latents are more consistent, *i.e.*, less jittery compared to non-aligned latents.

## 3.2 SPATIOTEMPORAL DIFFUSION TRANSFORMER

We extend the pretrained diffusion transformer architecture used in Step1X-3D (Li et al., 2025a) and Hunyuan3D (TencentHunyuan3DTeam, 2025a), originally designed to generate a single 3D latent shape from a single image, to instead produce a sequence of latents representing dynamic shapes inferred from a video. The decision to leverage a pretrained shape diffusion model is that it was trained on a much larger dataset than the available 4D data. This provides a valuable source for achieving generalization. To adapt the 3D model to the dynamic setting, we introduce temporal extensions that enable consistent generation across time.

The pretrained rectified-flow diffusion transformer (Li et al., 2025a) takes a conditioning image embedded with DinoV2 (Oquab et al., 2024), then applies a sequence of dual-stream and single-stream transformer blocks (BlackForestLabs, 2024). These blocks jointly attend to the hidden states of the image features and the noised shape latents. In the following, we propose two extensions.

**Inserting spatiotemporal attention layers.** Motivated by prior extensions of text-to-image diffusion models to video (Guo et al., 2023; Blattmann et al., 2023), we insert spatiotemporal transformer layers after each block of the pretrained model to capture temporal dependencies. These layers mirror the base model's single-stream blocks, but their self-attention operates jointly across shape and image hidden states from all frames. We also explored alternative variants, including attending only to shape hidden states and applying 1D temporal attention that excludes same-frame interactions. Empirically, both alternatives degraded generation quality. To encode frame indices, we add 1D RoPE embeddings (Su et al., 2024) to the hidden states. During training, only the spatiotemporal layers are updated, while the base model remains frozen to avoid forgetting, given the limited 4D data. Each spatiotemporal layer's output projection is zero-initialized to stabilize training.

**Sharing noise across frames.** In diffusion models, the additive Gaussian noise is sampled independently. However, in our setup, independent noise across frames leads to unstable motion. We suspect that this instability occurs because the base model is trained to generate 3D shapes without regard to the viewpoint. As a result, different noise samples push the model toward different poses and scales, causing visible flickering between frames.

In contrast, image and video diffusion models work on grid-structured data where each hidden state has explicit positional embeddings. These embeddings give attention layers exact spatiotemporal coordinates, so independent noise can be used per frame without breaking temporal consistency. By comparison, 3DShape2VectSet-style models operate on irregular structures without explicit position embeddings. They must infer position implicitly, which makes them more sensitive to variations in the noise. Inspired by practices in early image-to-video diffusion (Ge et al., 2023) and video-to-video editing (Burgert et al., 2025), we enforce temporal correspondence by replicating the same noise across frames during both training and inference. This simple yet effective strategy substantially improves temporal smoothness, yielding more consistently aligned shapes even before additional training. A visual comparison is provided in Fig. 4.

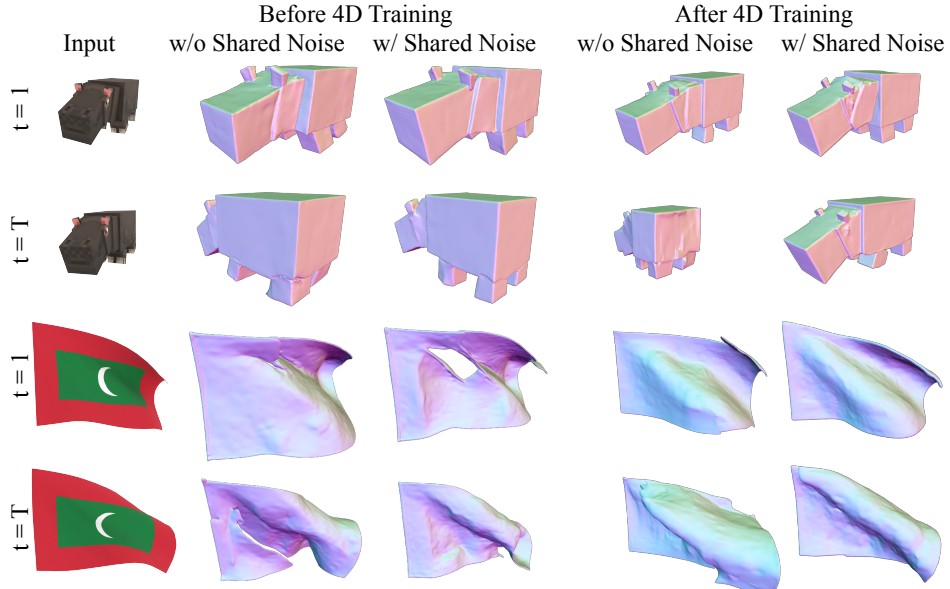

Figure 4: Qualitative comparison of noise sharing. The base 3D model generates object shapes in arbitrary orientations agnostic to the input image viewpoint, often causing pose changes across a sequence (*e.g.* the hippo in the first row). We observe that sharing noise across frames reduces flickering and further improves shape quality in challenging cases such as the flag example.

## 4 MESH REGISTRATION AND TEXTURIZATION

Once the sequence of meshes is generated, we apply a two-stage pipeline to ensure alignment with the input video and to produce consistent textures. This process consists of global pose registration and global texturization.

**Global pose registration.** Our mesh generation network is trained on animated objects in a canonical coordinate system, *i.e.*, with object geometries aligned to the coordinate axes. Consequently, the generated 4D geometry may not match the pose observed in the input video if the latter is not canonicalized. To address this, we adopt the pose registration strategy from V2M4 (Chen et al., 2025) to re-pose the generated 4D geometry so that it aligns with the input image. Owing to the stability and temporal consistency of our generated 4D geometry, we only need to estimate the pose for the first frame and then apply the same transformation globally to all subsequent frames, which achieves effective alignment with the input video.

**Global texturization.** A straightforward approach to texturizing 4D geometry is to apply off-the-shelf 3D mesh texturization methods (TencentHunyuan3DTeam, 2025a; Li et al., 2025a) independently to each frame. However, this often leads to inconsistency across frames due to the randomness in texture generation, especially in occluded or unseen regions such as the object's backside. To overcome this, we employ the pairwise mesh registration strategy by Chen et al. (2025) to convert our 4D geometry into a topology-consistent 4D mesh. We then apply off-the-shelf texturization methods to the first-frame mesh and propagate the resulting texture across all frames by leveraging the consistent topology.

Further details of the texturization steps are provided in the Appendix.

## 5 EXPERIMENTS

### 5.1 IMPLEMENTATION DETAILS

**Data curation.** We curated a set of 14k high-quality animated 3D assets from Objaverse. Since our model builds on Step1X-3D (Li et al., 2025a) and Hunyuan3D-2.1 (TencentHunyuan3DTeam, 2025a), we process the data to match its distribution. Each mesh is converted into a watertight version to remove internal surfaces. Root-body motion is removed from the animation, and objects

Table 1: Quantitative comparison with baselines, evaluated on the held-out Objaverse test set.

| | Representaion | Feedforward | Chamfer↓ | IoU↑ | F-Score ↑ | Time ↓ |
|---|---|---|---|---|---|---|
| Step1X-3D (Li et al., 2025a) | SDF | ✓ | 0.1356 | 0.3033 | 0.2617 | 3 min |
| Step1X-3D + shared noise | SDF | ✓ | 0.1368 | 0.3149 | 0.2817 | 3 min |
| DreamMesh4D (Li et al., 2024c) | mesh | ✗ | 0.2108 | - | 0.2422 | 40 min |
| L4GM (Ren et al., 2024) | MV-3D GS | ✓ | 0.1576 | – | 0.1932 | 25 sec |
| V2M4 (Chen et al., 2025) | mesh + deform | ✗ | 0.1233 | 0.3023 | 0.2814 | 30 min |
| GVFD (Zhang et al., 2025) | 3D GS + deform | ✓ | 0.3978 | – | 0.0699 | 10 min |
| ShapeGen4D (Step1X-3D) | SDF | ✓ | 0.1220 | 0.3276 | 0.2934 | 3 min |
| ShapeGen4D (Hunyuan3D-2.1) | SDF | ✓ | **0.0827** | **0.4155** | **0.3971** | 15 min |

are rescaled so that all shapes in a sequence are centered within a unit bounding box. We retain the original non-watertight mesh, where the animation is defined, to extract aligned query points for encoding, but filter out inner vertices based on their projected distance to the watertight mesh. For each sample, fixed-view videos are rendered from randomly sampled azimuth angles. To ensure consistent object size, videos are rendered with an $\alpha$ channel and rescaled so that the object occupies a fixed portion of the image.

**Training details.** The diffusion model is trained to generate 16 frames, each represented by 1024 latents. For every frame, we sample a 32k-point cloud from the watertight mesh as input to the encoder, paired with 1024 query points from the animated non-watertight mesh. Training is conducted with a batch size of 64 across 16 A100 GPUs, using a learning rate of $5 \times 10^{-5}$. We train for 25k iterations ( $\approx 2$ days ) after which the validation loss indicates negligible quality improvement.

## 5.2 ANALYSIS AND COMPARISONS

**Test datasets.** We collect two evaluation sets: (1) a held-out set of 33 animated samples from Objaverse, each with significant object motion and high-quality textures, used to evaluate geometric accuracy across methods; and (2) 20 video sequences from Consistent4D (Jiang et al., 2024b), covering both in-the-wild and synthetic subjects. Since ground-truth geometry is unavailable in the latter, we evaluate using rendering-based metrics instead.

**Baselines.** We conduct a comparison with state-of-the-art feedforward 4D generation methods, *i.e.*, *L4GM* (Ren et al., 2024) and *GVFD* (Zhang et al., 2025), both of which output Gaussian particles. For geometric evaluation, we threshold particle densities to retain only those near the surface and treat the particle centers as points. Since the resulting point clouds exhibit noisy surface artifacts, converting them into watertight meshes for IoU evaluation is technically challenging and therefore omitted. As a reference baseline, we also evaluate *Step1X-3D* applied independently to each frame.

**Metrics.** *Geometric metrics:* Since Step1X-3D and GVFD do not generate shapes aligned with the input video frames, pose registration is required for comparison with ground-truth geometry. We first estimate a global similarity transformation from the first frame of each sequence by grid-searching over azimuth angles, scaling, and translation (based on truncated Chamfer distance), followed by iterative closest point (ICP). This transformation is applied to the full sequence, with additional per-frame translations to ensure alignment while preserving original frame-wise rotation and scale changes from the prediction, allowing temporal consistency to be properly evaluated. We then follow Zhang et al. (2023) to report Chamfer distance and F-Score between point clouds, and IoU on occupancy voxel grids at $128^3$ resolution.

*Rendering metrics:* We render the textured shapes from the target camera view and evaluate with rendering metrics, including LPIPS (Zhang et al., 2018) and DreamSim (Fu et al., 2023) for perceptual similarity, CLIP score (Radford et al., 2021) for conceptual alignment between text and rendered images, and FVD (Unterthiner et al., 2019) for video quality.

**Quantitative comparison.** *On geometry:* As shown in Tab. 1, our method achieves more precise geometry compared to the baselines as indicated by the consistent gain across all three metrics.

*On rendering:* We show results in Tab. 2 on the Consistent4D dataset. Our method produces more consistent results than other generative methods, Step1X-3D and GVFD, yielding noticeably better renderings. Interestingly, L4GM still achieves higher scores, despite its geometry being qualitatively worse than ours. This advantage comes from its strong bias towards reconstructing the input views

Table 2: Quantitative comparison of rendering quality. L4GM has a unique advantage because its predictions are inherently aligned with the input image views, whereas other methods are not. Refer to Fig. 6 in the Appendix for visual comparisons.

|  | Aligned | LPIPS↓ | CLIP↑ | FVD↓ | DreamSim↓ |
|---|---|---|---|---|---|
| Step1X-3D | ✗ | 0.1524 | 0.9040 | 940 | 0.1106 |
| L4GM | ✓ | **0.0988** | **0.9397** | **302** | **0.0487** |
| GVFD | ✗ | 0.1691 | 0.8601 | 916 | 0.1467 |
| Ours | ✗ | 0.1359 | 0.9009 | 796 | 0.0966 |

Table 3: Ablation study by removing one component at a time.

|  | Chamfer↓ | IoU↑ | F-Score ↑ |
|---|---|---|---|
| w/o aligned latents | 0.1348 | 0.3230 | 0.3002 |
| w/o shared noise | 0.1186 | 0.3137 | 0.2962 |
| 1D temp. attn. | 0.2118 | 0.1503 | 0.1462 |
| w/o image hidden states | 0.1196 | 0.3332 | 0.3084 |
| w/o time shift | 0.1374 | 0.3087 | 0.2861 |
| Full method | **0.1096** | **0.3346** | **0.3190** |

at the expense of generating meaningful 4D shapes that also look plausible in other views. L4GM's outputs align closely with input viewpoints, reducing all metrics (LPIPS, DreamSim, CLIP, FVD). In contrast, Step1X-3D, GVFD, and our method are disadvantaged by camera-coordinate misalignment and the fact that they balance reconstruction of input views with generating meaningful 4D shapes that are also valid in other views.

**Qualitative comparison.** Fig. 5 and Fig. 6 further shows the limitation of baseline approaches. L4GM suffers from two main issues: (1) its multi-view image-based representation yields Gaussian particles that do not seamlessly merge, leading to imperfect geometry and ghosting artifacts in renderings; and (2) due to its limited model scale, it often produces incorrect shapes on challenging test cases. GVFD, on the other hand, generates jittery motions where objects distort significantly and fails to capture fine surface dynamics, such as the motion of a flag. Step1X-3D applied independently to each frame produces reasonable per-frame geometry but lacks temporal consistency, with poses drifting across frames. Using shared noise per sequence with Step1X-3D reduces global pose jitter but still results in noticeable shape fluctuations. In contrast, our method produces high-quality meshes, maintains consistent poses, and exhibits substantially less temporal jitter.

**Ablations.** We analyze each design choice through controlled ablation experiments by removing one component at a time from the full method. To reduce training cost, we use 8 frames instead of 16. Results on our held-out Objaverse dataset are reported in Tab. 3.

*Aligned latents:* We find that learning with aligned latents, obtained by using query points from the animated original non-watertight meshes, plays a central role. Replacing them with independently sampled query points per frame leads to quality degradation and more flickering.

*Shared noise:* We find that applying shared noise across frames noticeably reduces flickering and shape artifacts compared to independent noise. Visual comparisons are shown in Fig. 4. Without noise sharing, Step1X-3D (before training) produces random orientations of the hippo; even after training with spatiotemporal attention, it often yields inconsistent poses. By sharing noise, our model resolves this issue. Furthermore, we observe that training with shared noise can also improve geometry in challenging cases, such as the flag example, compared to baselines.

*Spatiotemporal attention architecture:* We explore two alternative architectural designs for spatiotemporal attention. (1) Instead of applying self-attention across all hidden states, we restrict it to 1D temporal attention: each hidden state only attends to the sequence of the same query point, without attending to other states within the same frame. While this design is common in video generative models (Guo et al., 2023), we find it leads to catastrophic failure. We hypothesize that, unlike image or video models where hidden states are explicitly embedded with spatial coordinates, our latent representation lacks such information; thus, intra-frame attention is necessary for hidden states to infer their spatial positions. (2) We also remove the image hidden states from the spatiotemporal attention layer and observe a performance drop, indicating that cross-frame dependencies among image features contribute positively.

*Denoise time shift:* With more latents and shared noise, the 4D diffusion transformer faces reduced prediction difficulty at the same noise level compared to 3D generation. Similar to strategies used in training multi-resolution image diffusion models (Esser et al., 2024), we apply a time shift to the denoising scheduler to allocate more steps at mid-to-high noise levels. We find that applying a time shift during denoising significantly improves the stability of our results, though training with a time shift has little effect.

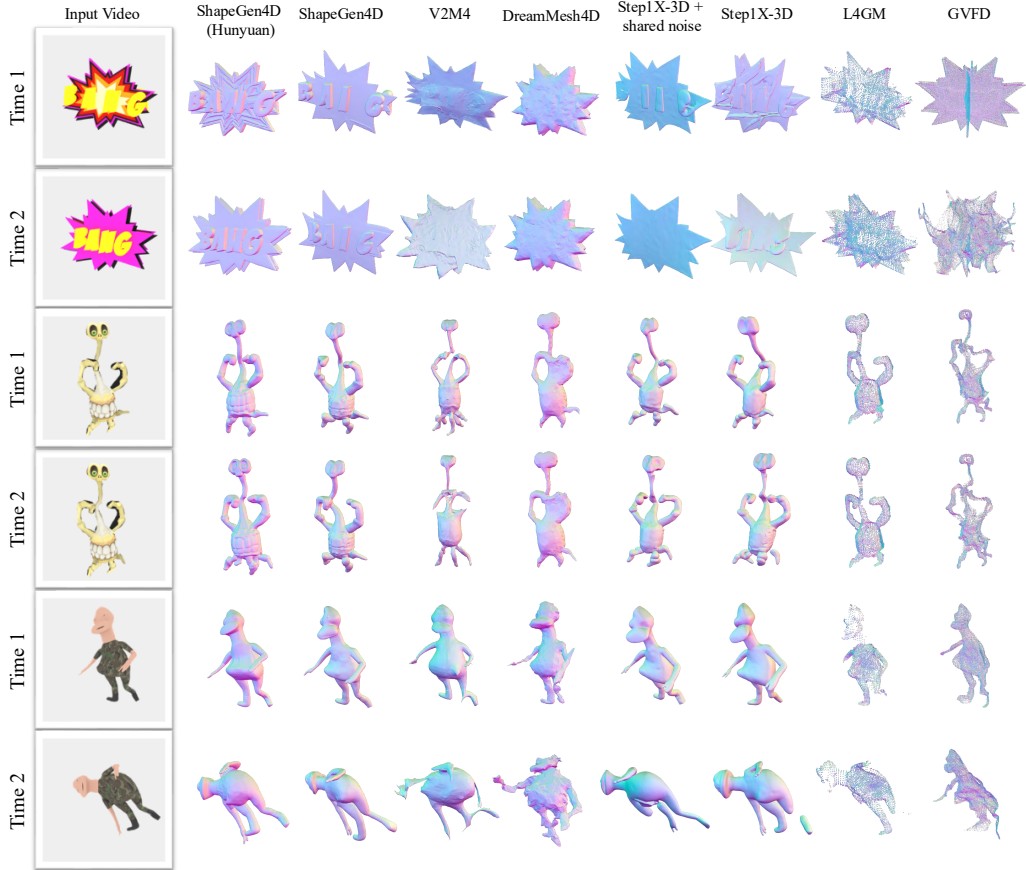

Figure 5: Qualitative comparison with baselines on the held-out Objaverse test set.

## 6 CONCLUSION

We propose a feedforward model that generates a sequence of temporally consistent 3D shapes from an input video. Our model builds on state-of-the-art 3D generation methods, extending them with spatiotemporal attention layers and careful design choices that preserve strong priors while enabling training with relatively small 4D datasets. Compared to recent baselines such as L4GM and GVFD, our approach demonstrates stronger generalization and produces higher-quality geometry than their 3D Gaussian-splatting representations.

Nonetheless, our method can be further improved with the following **limitations** in mind: (a) Inherited from the base 3D generation model, our framework is agnostic to the viewpoint of input video frames. As a result, it struggles to capture global motions such as rigid object rotations. Viewpoint-aligned reconstruction requires additional pose registration, which might be mitigated by finetuning with paired, viewpoint-aligned images and 3D shapes. (b) To create fully animatable textured assets, additional steps such as pose registration and texture propagation are still necessary. (c) Local temporal jitter remains visible in some results. Following lessons from video diffusion, we believe a spatiotemporal 3D VAE may help reduce flickering.

### ACKNOWLEDGMENTS

We thank Avalon Vinella for assistance in preparing the Objaverse dataset. We are also grateful to Willi Menapace for his advice on training flow matching. Finally, we thank Gleb Dmukhin, Hao Zhang, Michael Vasilkovsky, Sergei Korolev, and Vladislav Shakhray for their valuable discussions and support.

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

APPENDIX

## A  MESH REGISTRATION AND TEXTURIZATION DETAILS

In this section, we present the detailed implementations of mesh registration and texturization.

Let the input video sequence of length $T$ be denoted as $V_{\text{ref}} = \{V_{\text{ref},t}\}_{t=1}^T$, and let the generated untextured 4D geometry be $\widetilde{\mathcal{M}}_{\text{init}} = \{\widetilde{\mathcal{M}}_1, \widetilde{\mathcal{M}}_2, \ldots, \widetilde{\mathcal{M}}_T\}$. We align these meshes to the poses in the reference video and subsequently apply texturization through a two-step process: *Global Pose Registration* and *Global Texturization*, inspired by V2M4 (Chen et al., 2025).

### A.1  GLOBAL POSE REGISTRATION

To align the generated geometry with the video input, we reformulate the task from mesh pose estimation to camera pose estimation. The key idea is that if we can determine the camera pose under which a rendered view is semantically similar to the corresponding video frame, then by inverting this pose and applying the transformation to the geometry, we can register the mesh to the input video pose.

We parameterize the camera pose as $\mathcal{C} \in \mathbb{R}^6 = \{\text{yaw}, \text{pitch}, \text{radius}, \text{lookat}_{x,y,z}\}$ for each untextured mesh $\widetilde{\mathcal{M}}_t$. The camera pose estimation procedure consists of three stages:

**1) Candidate sampling.** We first sample a large number of candidate camera positions around the object and select the top $n$ positions that yield the highest semantic similarity—measured by DreamSim (Fu et al., 2023)—to the reference video frame $V_{\text{ref},t}$.

**2) Pose prediction via VGGT.** We then employ VGGT (Wang et al., 2025c) to process the rendered views from these $n$ candidate cameras together with the reference frame $V_{\text{ref},t}$, obtaining predicted point clouds for each view. Let $\overline{\text{PC}}_{\text{ref}}$ and $\overline{\text{PC}}_{\text{rend},\{1,\ldots,n\}}$ denote the predicted point clouds for $V_{\text{ref},t}$ and $V_{\text{rend},\{1,\ldots,n\}}$, respectively. Since the camera models, poses, and mesh $\widetilde{\mathcal{M}}_t$ used for rendering $V_{\text{rend},\{1,\ldots,n\}}$ are known, we can compute the corresponding ground-truth point clouds in our coordinate system, denoted as $\text{PC}_{\text{rend},\{1,\ldots,n\}}$. We align the predicted point clouds with the ground truth using a global similarity transformation (rotation, translation, and scale) optimized via the Chamfer distance. Applying this transformation to $\overline{\text{PC}}_{\text{ref}}$ yields $\text{PC}_{\text{ref}}$. Ideally, under the correct camera pose $\mathcal{C}_{\text{ref},t}$, the 3D coordinates of $\text{PC}_{\text{ref}}$ should project consistently onto the 2D pixel coordinates of $V_{\text{ref},t}$. We therefore optimize the camera extrinsic parameters to enforce this projection, resulting in the initial estimate $\mathcal{C}_{\text{ref},t,\text{VGGT}}$.

This estimated camera pose is then added to the pool of candidate poses obtained in step 1. We further refine it using Particle Swarm Optimization (PSO), which searches for a more accurate pose $\mathcal{C}_{\text{ref},t,\text{PSO}}$ by comparing the rendered images with the reference frame.

**3) Pose refinement via gradient descent.** Starting from $\mathcal{C}_{\text{ref},t,\text{PSO}}$, we further refine the camera pose through gradient-based optimization. Specifically, we minimize the discrepancy between the mask of the rendered view—defined as the pixel region occupied by the object during rasterization—and the foreground region of $V_{\text{ref},t}$. This yields the final camera pose estimate $\mathcal{C}_{\text{ref},t}$.

Once $\mathcal{C}_{\text{ref},t}$ is determined, we compute the corresponding camera motion, invert it, and apply the transformation to $\widetilde{\mathcal{M}}_t$, obtaining the pose-registered mesh $\widetilde{\mathcal{M}}_{\text{repose},t}$.

### A.2  GLOBAL TEXTURIZATION

As discussed in the main paper, independently texturizing each frame's geometry leads to appearance inconsistencies due to topology variations across frames. To address this, we first convert the 4D geometry into a topology-consistent representation and then perform texturization.

**1) Topology-consistent geometry conversion.** For each pair of consecutive meshes along the time dimension, $\widetilde{\mathcal{M}}_t$ and $\widetilde{\mathcal{M}}_{t+1}$, we treat $\widetilde{\mathcal{M}}_t$ as a rigid body. We first optimize its global transformation (rotation, translation, and scale) to align it with $\widetilde{\mathcal{M}}_{t+1}$ using a combination of Chamfer distance and differentiable rendering loss. After achieving global registration, we further refine the alignment

| Method | Chamfer↓ | IoU↑ |
|--------|----------|------|
| KVQ (watertight) | 0.0246 | 0.8680 |
| Q (original) | 0.0369 | 0.8401 |

Table 4: Watertight vs. original query points.

| Method | Chamfer↓ | IoU↑ | F-Score↑ |
|--------|----------|------|----------|
| Baseline | 0.0929 | 0.3739 | 0.3557 |
| MultiDiffusion | 0.0934 | 0.3779 | 0.3499 |

Table 5: Baseline vs. MultiDiffusion.

using the preconditioned optimization method of Nicolet et al. (2021), which enables fast and stable convergence during the inverse reconstruction process. To enhance local registration, we also apply the As-Rigid-As-Possible (ARAP) constraint in conjunction with the Chamfer distance.

Through these global and local alignment steps, we obtain a deformed version $\widetilde{\mathcal{M}}'_t$, with vertices $\mathcal{V}'_t$ and faces $\mathcal{F}'_t$, that shares identical topology ($\mathcal{V}'_t = \mathcal{V}_t$, $\mathcal{F}'_t = \mathcal{F}_t$) while closely matching the shape of $\widetilde{\mathcal{M}}_{t+1}$. We then replace the original $\widetilde{\mathcal{M}}_{t+1}$ with the deformed mesh $\widetilde{\mathcal{M}}'_t$. In practice, we start at $\widetilde{\mathcal{M}}_1$ and iteratively apply this registration process across consecutive frames, yielding a sequence of topology-consistent meshes $\widetilde{\mathcal{M}}'_{\{1,...,T\}}$.

**2) Texturization.** After obtaining topology-consistent geometry, we first apply an existing texture generation method to texturize the initial mesh $\widetilde{\mathcal{M}}'_1$. The resulting texture map and UV coordinates are then directly reused for subsequent meshes $\widetilde{\mathcal{M}}'_{\{2,...,T\}}$, ensuring appearance consistency across the entire sequence.

## B    ADDITIONAL RESULTS

Fig. 6 visualizes the colored reconstructions from each method. Our approach produces the most temporally consistent shapes and captures fine-grained motion details from the input images. Step1X-3D can capture motion but suffers from jitter and random object poses. GVFD often fails to model motion, highlighting its limited ability to generalize deformations. L4GM generates high-quality results in some cases (e.g., cats), but fails catastrophically on challenging examples such as water splashes, where misfused 3D Gaussian splats lead to ghosting artifacts.

## C    TOPOLOGY CHANGE RESULTS

Figure 7 shows results on topology changes which are hardly modeled by using 1 mesh and deformation field. Each group shows three time steps (rows) for one sequence, with the input images on the left and our generated meshes rendered on the right. The examples are as follows: emerging parts (e.g., the appearance of a gun barrel on a pistol), object fusion and splitting (e.g., an egg merging into a cup), tearing events (e.g., a shirt ripping), morphing (e.g., a Lego brick transforming into a cat), growing objects (e.g., a flower blooming), and object shattering (e.g., the explosion of a gift box).

## D    VAE RESULTS

Both Step1X-3D and the Hunyuan VAE are trained on watertight meshes. A concern is that sampling query points directly on the original meshes might degrade performance or introduce artifacts. We evaluate this in Table 4 where we can see marginal degradation in reconstruction quality compared to sampling points from the watertight mesh. Upon visual inspection, there is no noticeable difference in the reconstructed surfaces as seen in Fig 8. We hypothesize that this discrepancy may be due to small surface offsets introduced by the different sources of query points.

## E    MULTIDIFFUSION RESULTS

Here, we show how to extend our method to longer videos by leveraging MultiDiffusion (Bar-Tal et al., 2023) techniques from the long video generation literature. MultiDiffusion [1], a commonly

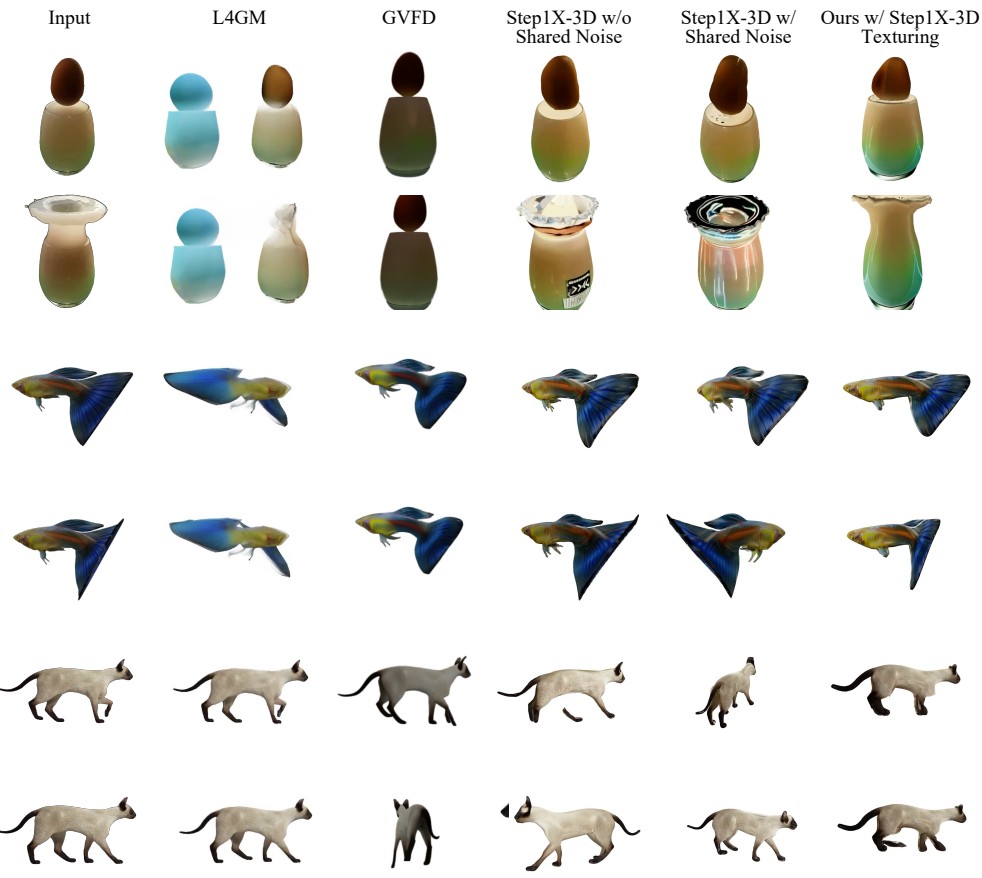

Figure 6: Qualitative comparison on Consistent4D test set.

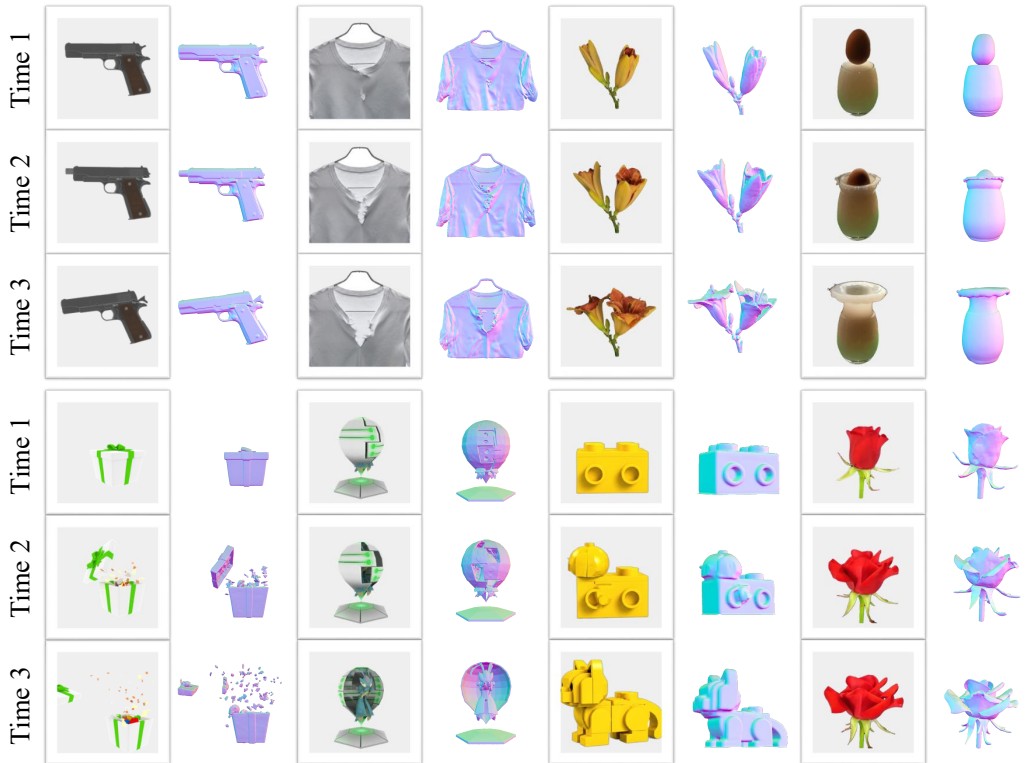

Figure 7: Topology Change Results. Our method can handle topology changes: emerging parts, tearing, growth, fusion and splitting, morphing, and object shattering. Each column shows the input view (left) and our reconstructed mesh (right) across three time steps.

used technique for panoramic image or long video generation. The method applies the 16-frame diffusion model in a sliding-window manner: at each diffusion step, noise predictions are computed for all overlapping windows in parallel, and the predictions in the overlapping regions are fused by averaging. We note that MultiDiffusion is theoretically suboptimal, as information propagation between distant temporal windows is inefficient and may result in reduced global consistency. In Table 5, we see a small drop in performance when using MultiDiffusion. Visual results can be seen in Figure 9.

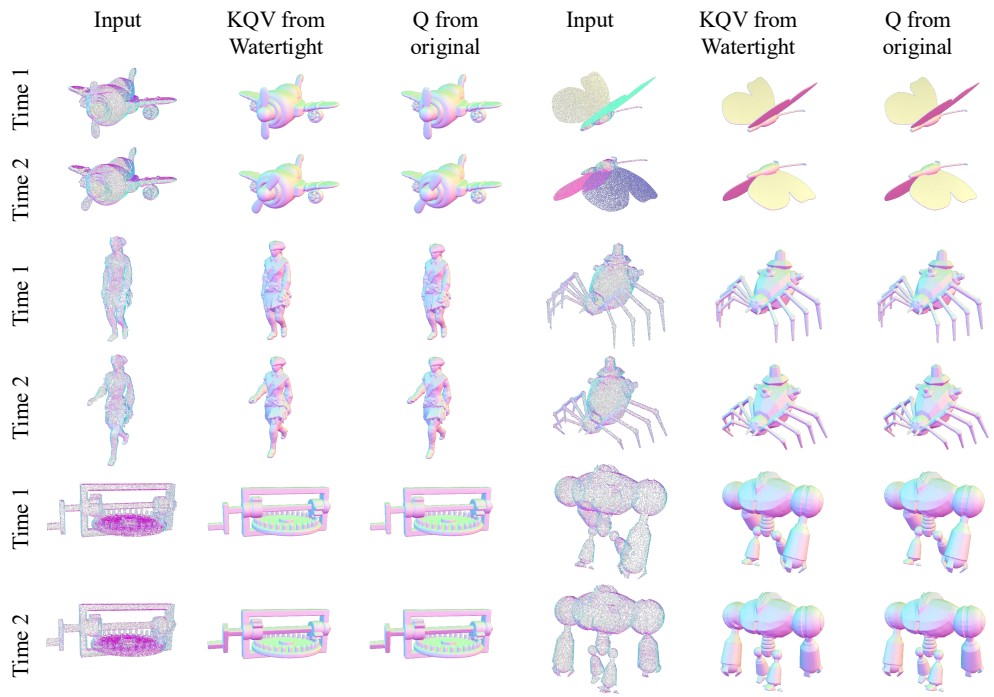

Figure 8: Qualitative comparison of VAE reconstructions using query points from the watertight mesh versus the original mesh. The first column shows the input point clouds, while the second and third columns show the reconstructed surfaces when using KQV from the watertight mesh and Q directly from the original mesh. Both settings produce visually similar geometry across time.

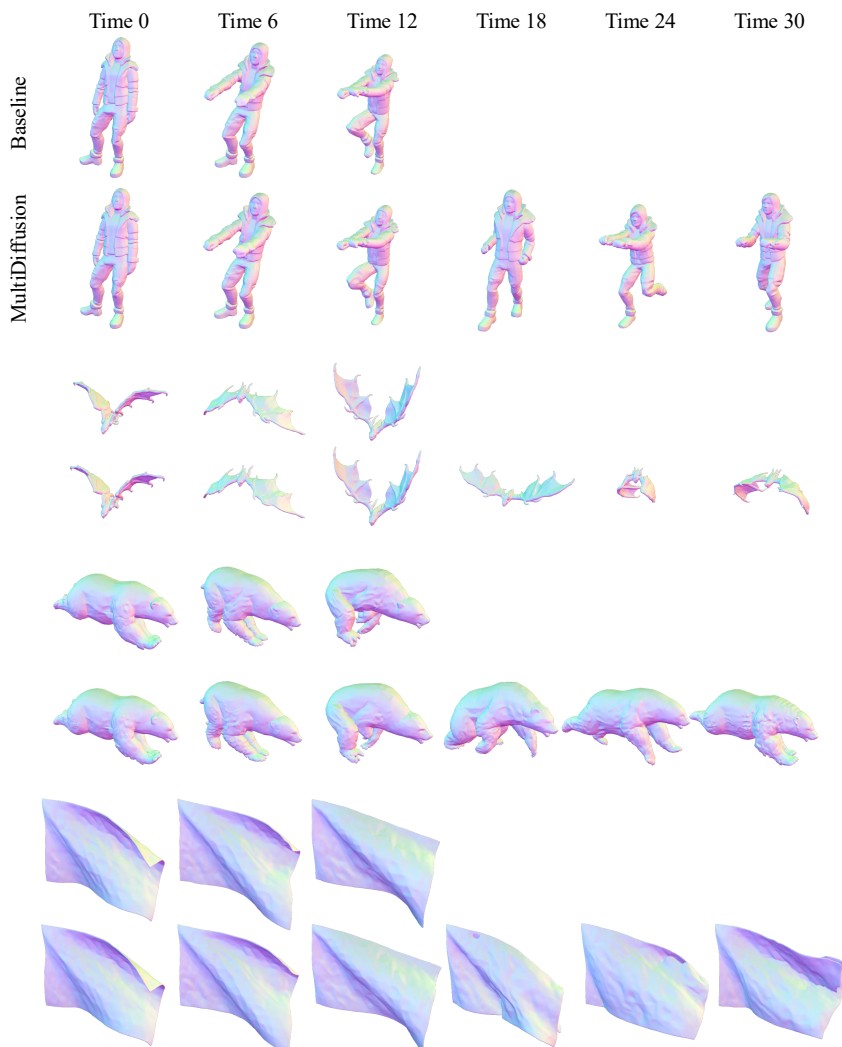

Figure 9: Qualitative comparison of our baseline vs MultiDiffusion. Each column is a time step sampled from each sequence. The baseline is evaluated on frames 0–16, while MultiDiffusion extends the sequence to frames 0–32 using overlapping windows. Despite being trained only on 16-frame videos, it produces visually coherence and temporally consistent geometry over the longer sequence.

