# OpenReview forum: "ShapeGen4D: Towards High Quality 4D Shape Generation from Videos"
_ICLR.cc/2026/Conference — ICLR 2026 Poster_

### Official Review · Reviewer_6RuC · 2025-10-28

**Soundness:** 3
**Presentation:** 3
**Contribution:** 3
**Rating:** 6
**Confidence:** 4

**Summary:**

This paper introduces ShapeGen4D, a feedforward framework for high-quality 4D shape generation from monocular videos. The method builds upon a pre-trained 3D generative model (Step1X-3D) and adapts it to the dynamic setting through three key innovations:

1. Spatiotemporal attention layers to capture cross-frame dependencies;
2. Temporally-aligned latent encoding to reduce jitter and improve consistency;
3. Shared noise across frames to enhance temporal stability.

ShapeGen4D directly outputs a sequence of 3D meshes, supporting non-rigid motion, volume changes, and topological transitions without per-frame optimization.

**Strengths:**

1. End-to-end 4d generation framework. The paper presents a direct, feedforward approach for generating 4D mesh sequences from a single video, which is a simplification over more complex optimization-based or multi-stage pipelines.
2. Handling of complex dynamics. The framework demonstrates the capability to generate a range of dynamic phenomena, including non-rigid motion and topological changes, which are challenging for methods restricted to simpler deformations.
3. Clear written. The paper is clearly written and provides a well-organized summary of related work, helping to situate the contributions within the field of 4D generation.

**Weaknesses:**

1. Inherent Limitations in Temporal Geometry Consistency. While the proposed techniques of latent alignment and noise sharing effectively reduce temporal jitter, they do not establish an explicit, parametric model of motion (e.g., a deformation field). The framework still generates each frame's mesh independently from a sequence of latents. This inherently discrete representation may struggle to guarantee as-rigid-as-possible or physically plausible transitions over time, potentially leading to subtle topological inconsistencies or unnatural deformations that are not explicitly regularized. This limitation is indirectly acknowledged by the authors, who note that "local temporal jitter remains visible in some results."
2. Limited Scalability to Long Video Sequences. The model is designed to generate a fixed-length sequence (e.g., 16 frames) in a single forward pass, constrained by the memory and architecture of the underlying diffusion transformer. This fixed-horizon generation prevents the method from processing arbitrarily long videos, a common requirement in real-world applications. The paper does not explore mechanisms for temporal auto-regressive generation or sliding-window inference, which would be necessary to scale to longer durations, potentially at the cost of error accumulation across segments.

**Questions:**

1. Since the meshes for consecutive timesteps are generated independently from discrete latents, the resulting 4D sequence lacks an explicit, continuous deformation field. This may lead to non-smooth interpolations and visually incoherent dynamics when rendered at frame rates higher than the generation rate. Do you have plans to incorporate a post-processing step or an intermediate representation (e.g., a neural deformation field) to enable truly continuous, smooth morphing between the generated key meshes? How might this be integrated into your current pipeline?
2. The current framework generates a fixed number of frames (e.g., 16) in a single feedforward pass. What is the potential of extending ShapeGen4D to handle arbitrarily long videos? For instance, could an autoregressive approach be adopted, where the generation of a subsequent clip is conditioned on the final frames of the previous clip, similar to the strategy employed by L4GM? If so, what would be the main technical challenges, such as error accumulation or maintaining global consistency across segments?

---

> ### Author Response · Authors · 2025-11-22
>
> ### [W1] No explicit motion modeling, how to get deformation field?
> That is great question. To answer this, we would like to first explain why we prefer an implicit modeling. Then we explain how to obtain deformation.
>
>
>
>
> **(i) Implicit modeling is more scalable:** First, we do not view not explicitly modeling motion as a limitation; rather, we see it as a promising property for achieving more generalizable 4D generation in the future. We draw inspiration from the evolution of image-to-video generation. Early approaches relied on hand-crafted, explicit motion heuristics such as pixel warping, which limited both robustness and generalization. In contrast, recent state-of-the-art video generation models no longer explicitly model motion. Instead, they learn it implicitly and can generate visually coherent and temporally consistent content under complex scenarios without noticeable flickering.
>
> We adopt the same philosophy. A pipeline without explicit motion modeling not only simplifies the formulation but also fundamentally improves scalability to arbitrary scene types. For instance, it enables handling complex topological changes such as objects with emerging parts (e.g., the appearance of a gun barrel on a pistol) or object fusion and splitting (e.g., an egg merging into a cup)—phenomena that are inherently difficult to capture using models with explicit motion prediction (e.g., generating position offsets as in GVFD).
>
> Although our current results are not perfect and still exhibit minor jittering, we believe that with larger-scale training, improved VAE architectures, and moderately expanded datasets, 4D generation can reach a level of quality comparable to recent advances in video generation. In fact, we found that replacing the base model from Step1X3D (used in the original draft) with a more powerful backbone, such as Hunyuan3D-2.1, noticeably reduced flickering artifacts, as demonstrated in the updated supplementary video. We have included visualizations of the updated ShapeGen4 results based on Hunyuan3D in **Figure 5**, as well as quantitative evaluations in **Table 1** (also shown in the table below), which demonstrate the improvements achieved by switching the base model.
>
> | Method                 | Chamfer↓ | IoU↑                 | F-Score↑ |
> |------------------------|--------------|----------------------|---------|
> | ShapeGen4D (Step1X-3D)            | 0.1220       | 0.3276              | 0.2934  |
> | ShapeGen4D (Hunyuan 2.1) | 0.0827 |  0.4155|  0.3971|
>
> **(ii) How to obtain explicit deformation:** Although our model does not directly predict deformation fields, it produces a sequence of 3D meshes from which motion can be derived. Estimating registration between these meshes as a post-processing step is relatively a simpler task compared to directly generating deformations from scratch as in GVFD. For instance, one can apply the optimization-based registration procedure described in Section 4 to obtain continuous motion interpolation if desired.
>
> To improve motion smoothness, we followed V2M4 and adopt commonly used physics-constrained losses such as the as-rigid-as-possible regularization. However, we acknowledge that current optimization-based registration methods are not perfect since overweighting the regularization terms may lead to over-smoothed or distorted motion. In the longer term, we suggest exploring the training of feedforward models to estimate deformations directly from mesh sequences or pairs, with the goal of achieving robustness comparable to optical flow estimation networks.

---

> > ### Author Response · Authors · 2025-11-22
> >
> > ### [W2, Q2] How to handle longer videos? Could an autoregressive approach be adopted?
> >
> > That is an great comment. Yes, an autoregressive approach can indeed be adopted to extend the temporal duration. However, given the recent progress in long video generation, more sophisticated strategies can be employed beyond the basic formulation in L4GM to mitigate error accumulation (i.e., exposure bias) and maintain global temporal consistency.
> >
> > We can leverage existing techniques from the long video generation literature to scale our model for longer sequences, and several approaches can be utilized for this purpose.
> >
> > In the updated draft, we present results of generating 32 frames using **MultiDiffusion** [1], a widely used technique for panoramic image and long video generation. This method applies the 16-frame diffusion model in a sliding-window fashion: at each diffusion step, noise predictions are computed for all overlapping windows in parallel, and the predictions in the overlapping regions are fused by averaging. However, MultiDiffusion is theoretically suboptimal, as information propagation between distant temporal windows is inefficient and may lead to reduced global consistency. Additional visual comparisons are provided in **Figure 9** of the appendix. Our quantitative evaluation shows no noticeable quality degradation caused by multi-diffusion, as presented in the table below.
> >
> > | Method               | Chamfer↓          | IoU ↑                     | F-score ↑                 |
> > |----------------------|-------------------------|----------------------------|----------------------------
> > | Apply 16-frame model on 16 frames    | 0.0929|0.3739   | 0.3557|
> > | Apply MultiDiffusion on 32 frames     |  0.0934  |  0.3779|  0.3499|
> >
> > A more promising direction lies in methods such as **Diffusion Forcing** [2] and **Self Forcing** [3], which distill diffusion models to generate frames autoregressively over long durations while effectively mitigating exposure bias. Moreover, in the context of video-grounded 4D generation, the issue of exposure bias—commonly observed in text-video generation—may be inherently less severe due to the temporal grounding provided by the input video.
> >
> > [1] MultiDiffusion: Fusing Diffusion Paths for Controlled Image Generation, ICML 2023
> >
> > [2] Diffusion Forcing: Next-token Prediction Meets Full-Sequence Diffusion, NeurIPS 2025
> >
> > [3] Self Forcing: Bridging the Train-Test Gap in Autoregressive Video Diffusion, arXiv 2025

---

> > > ### Comment · Reviewer_6RuC · 2025-11-24
> > > **Thanks for rebuttal**
> > >
> > > I'd like to thank the authors for their rebuttal. The authors provide an explanation based on the development of video generation models regarding the advantages of implicit modeling, which I find reasonable and insightful. The additional supplementary experiments further demonstrate that ShapeGen4D can be extended to handle longer videos, effectively addressing my concerns. Considering that the method itself is relatively simple and that similar ideas are already common in other domains (e.g., image generation → multi-view image generation, image generation → video generation), I decide to maintain my score.

---

### Official Review · Reviewer_pzzF · 2025-10-28

**Soundness:** 3
**Presentation:** 3
**Contribution:** 2
**Rating:** 6
**Confidence:** 4

**Summary:**

The paper introduces ShapeGen4D, a direct, feedforward framework for producing 4D shape sequences from monocular video input. The method extends an existing 3D shape diffusion model into the temporal domain through several architectural modifications. Firstly, the authors obtain temporally aligned shape latents through a time-aware point sampling strategy. Then, the pretrained shape DiT is finetuned with incorporation of spatiotemporal attention. During training, all shape frames share same noise to enhance temporal stability. Extensive experiments on diverse benchmarks demonstrate the strong qualitative and quantitative results.

**Strengths:**

* Conceptual originality: the first to adapt pretrained 3D shape diffusion generator for 4D shape sequence generation.
* End-to-end pipeline: the proposed method produces temporally coherent mesh sequences directly from video, avoiding costly optimization.
* Simple but effective modification: the model builds on a well-known 3D backbone with several straightforward and effective architectural modifications, achieving quantitative and qualitative improvements compared to baseline methods.

**Weaknesses:**

* Underlying gap with respect to pretrained prior: the queries of latents are sampled from non-watertight mesh, while the base 3D backbone is pretrained on watertight queries. Although the authors have explained the reason to do so (to avoid costly mesh registration), the potential performance drop still exists due to this gap.
* No explicit motion modeling: the generated mesh under each frame is independent with each other. The lack of explicit motion constraint cannot guarantee physically plausible, leading to minor jitter and limiting the application to continuous motion interpolation.
* Technically lack of novelty: the main components of this framework are mostly common practices in this field.

**Questions:**

* Current design only handles short clips (e.g., 16 frames), have the authors considered scaling the model to longer videos?

---

> ### Author Response · Authors · 2025-11-22
>
> ### [W1] Does performance drop when query latents are sampled from non-watertight meshes?
> That is a great question. Sampling query points directly from non-watertight meshes without any post-processing would lead to catastrophic failures in the VAE, since the original VAE model was never trained to handle points sampled from objects containing internal surfaces.
>
> To mitigate this issue, we adopt a procedure that avoids sampling points belonging to the internal structure of the object. Specifically, we first sample points from the watertight mesh and then project these points onto the original mesh. The projected points are subsequently used as query points.
>
> We find that encoding the points sampled through the above procedure yields marginal degradation in reconstruction accuracy compared to sampling points from the watertight mesh (see Table below). **However, visual inspection reveals no noticeable difference in the reconstructed surfaces.** We hypothesize that this discrepancy may be due to small surface offsets introduced by the different sources of query points. Additional visual comparisons are provided in **Figure 8** of the appendix.
>
> | Method               | Chamfer↓          | IoU ↑                     |
> |----------------------|-------------------------|----------------------------
> | KVQ (watertight)     | 0.0246 | 0.8680  |
> | Q (original)         | 0.0369   | 0.8401  |

---

> ### Author Response · Authors · 2025-11-22
>
> ### [W2] No explicit motion modeling
> We agree with the reviewer that our model does not explicitly model motion. However, we do not view this as a limitation; rather, we see it as a
> property for achieving more generalizable 4D generation in the future.
>
> **(i) Implicit modeling is more scalable:** We draw inspiration from the evolution of image-to-video generation. Early approaches relied on hand-crafted, explicit motion heuristics such as pixel warping, which limited both robustness and generalization. In contrast, recent state-of-the-art video generation models no longer explicitly model motion. Instead, they learn it implicitly and can generate visually coherent and temporally consistent content under complex scenarios without noticeable flickering.
>
> We adopt the same philosophy. A pipeline without explicit motion modeling not only simplifies the formulation but also fundamentally improves scalability to arbitrary scene types. For instance, it enables handling complex topological changes such as objects with emerging parts (e.g., the appearance of a gun barrel on a pistol) or object fusion and splitting (e.g., an egg merging into a cup)—phenomena that are inherently difficult to capture using models with explicit motion prediction (e.g., generating position offsets as in GVFD).
>
> **(ii) Explicit deformation can be acquired by postprocessing:** Moreover, although our model does not directly predict deformation fields, it produces a sequence of 3D meshes from which motion can be derived. Estimating registration between these meshes as a post-processing step is relatively simpler compared to directly generating deformations from scratch. For example, one can apply the registration procedure described in Section 4 to obtain continuous motion interpolation if desired.
>
> Finally, although our current results are not perfect and still exhibit minor jittering, we believe that with larger-scale training, improved VAE architectures, and moderately expanded datasets, 4D generation can reach a level of quality comparable to recent advances in video generation. In fact, we found that replacing the base model from Step1X3D (used in the original draft) with a more powerful backbone, such as Hunyuan3D-2.1, noticeably reduced flickering artifacts. We have included visualizations of the updated ShapeGen4 results based on Hunyuan3D in **Figure 5**, as well as quantitative evaluations in **Table 1** (also shown in the table below), which demonstrate the improvements achieved by switching the base model.
>
> | Method                 | Chamfer↓ | IoU↑                 | F-Score↑ |
> |------------------------|--------------|----------------------|---------|
> | ShapeGen4D (Step1X-3D)            | 0.1220       | 0.3276              | 0.2934  |
> | ShapeGen4D (Hunyuan 2.1) | 0.0827 |  0.4155|  0.3971|

---

> ### Author Response · Authors · 2025-11-22
>
> ### [Q1] How to handle longer videos?
> We can leverage existing techniques from the literature on long video generation to scale our model for longer video sequences, and several approaches can be adopted for this purpose.
>
> In the updated supplementary, we provide results of generating 32 frames using **MultiDiffusion** [1], a commonly used technique for panoramic image or long video generation. The method applies the 16-frame diffusion model in a sliding-window manner: at each diffusion step, noise predictions are computed for all overlapping windows in parallel, and the predictions in the overlapping regions are fused by averaging. We note that MultiDiffusion is theoretically suboptimal, as information propagation between distant temporal windows is inefficient and may result in reduced global consistency. Additional visual comparisons are provided in **Figure 9** of the appendix. Our quantitative evaluation shows no noticeable quality degradation caused by multi-diffusion, as presented in the table below.
>
> | Method               | Chamfer↓          | IoU ↑                     | F-score ↑                 |
> |----------------------|-------------------------|----------------------------|----------------------------
> | Apply 16-frame model on 16 frames    | 0.0929|0.3739   | 0.3557|
> | Apply MultiDiffusion on 32 frames     |  0.0934  |  0.3779|  0.3499|
>
> A more promising direction involves techniques such as **Diffusion Forcing** [2] or **Self Forcing** [3], which distills the model for autoregressive long-duration generation. Moreover, for video-grounded 4D generation, the challenge of exposure bias—prominent in long video generation—may be theoretically less severe due to the grounding provided by the input video.
>
> [1] MultiDiffusion: Fusing Diffusion Paths for Controlled Image Generation, ICML 2023
>
> [2] Diffusion Forcing: Next-token Prediction Meets Full-Sequence Diffusion, NeurIPS 2025
>
> [3] Self Forcing: Bridging the Train-Test Gap in Autoregressive Video Diffusion, arXiv 2025

---

### Official Review · Reviewer_uhdb · 2025-10-31

**Soundness:** 2
**Presentation:** 2
**Contribution:** 2
**Rating:** 4
**Confidence:** 5

**Summary:**

The paper works on 4d shape generation from monocular video. This is a new task and the authors aim to generate per-frame yet temporally coherence mesh instead of static mesh and deformation field to accommodates variantions in topology and relaxe constrains on the type of possible animations. Specifically, they construct the 4d shape generation model on a 3d shape generatoin work, and add temporal attention layers to ensure temporal consistency. Besides, they use global pose registration and global texturation as post-processing steps to better present the generated results and evaluation. Experiments on public and collected datasets validate the performance of the proposed model.

**Strengths:**

1) the generated shape seems to be of high-quality in terms of single-frame results, probably benifiting from high-quality pretrained 3D generation model weight

2) the paper is well-written and present its contribution in a clear way

3 )the authors work an new task, and propose a new pipeline of 4D shape generation

**Weaknesses:**

1) The generated results show noticeable flickering artifacts in both geometry and texture, where the texture flickering may stem from the instability in geometry.
WPOwpo
2)  While the authors claim that per-frame mesh generation is intended to capture variations in topology and enable a wider range of animations, the paper provides only a single example illustrating this capability (the BANG case in the supplementary material). The remaining examples appear to be rendered from skeleton-based animation models without topology changes and with a limited diversity of animation types. The authors are encouraged to present examples of topology-changing animations include: object shattering, characters growing extra limbs, soft-body fusion or splitting, cloth tearing, and morphing into an entirely different mesh structure.

3) Lack of comparison with dynamic mesh generation method, for example DreamMesh4D and DriveAnyMesh, both are video-4d mesh generation method. Besides, the authors are encouraged to compare the generation result with general video-4d method beyond L4GM.

**Questions:**

It is unclear why the proposed method does not incorporate data augmentation using geometric or spatial transformations during training, which might have alleviated or removed the requirement for global registration.
If my concerns are solved, I'll raise the recommendation.

---

> ### Author Response · Authors · 2025-11-22
>
> ### [W2] More examples of topology-changing animations
> We appreciate this suggestion. In the revision, we have added additional examples of topology-changing animations in the appendix (see Figure 7). These include objects with emerging parts (e.g., the appearance of a gun barrel on a pistol), examples of object fusion and splitting (e.g., an egg merging into a cup), tearing events (e.g., a shirt ripping), morphing (e.g., a Lego brick transforming into a cat), growing objects (e.g., a flower blooming), and object shattering (e.g., the explosion of a gift box).

---

> ### Author Response · Authors · 2025-11-22
>
> ### [W3] Compare to other dynamic mesh generation method
> We appreciate this valuable suggestion. In the original draft, we included comparisons with V2M4 (ICCV 2025), which has been shown to outperform DreamMesh4D. To address the reviewer’s concern, we have now added DreamMesh4D in our comparisons. As shown in **Table 1** in the revision and the table below, DreamMesh4D produces worse results than both V2M4 and our method. The generated geometry has worse quality as shown in the visualization shown in **Figure 5**. Furthermore, both V2M4 and DreamMesh4D are optimization-based approaches that require multiple processing steps and over half an hour to generate a single result. In contrast, our method produces 16 meshes **in only 3 minute**. Unfortunately, we could not perform a direct comparison with DriveAnyMesh, as its implementation is not publicly available. Finally, we updated our results using a stronger base model, Hunyuan 2.1, which provides better detail and improved alignment with the input images compared to Step1X-3D.
>
> | Method                 | Chamfer↓ | IoU↑                 | F-Score↑ | Time↓   |
> |------------------------|--------------|----------------------|---------|--------|
> | Step1X-3D              | 0.1356       | 0.3033              | 0.2617  | 3 min  |
> | Step1X-3D + shared noise  | 0.1368       | 0.3149              | 0.2817  | 3 min  |
> | DreamMesh4D            | 0.2108       | - | 0.2422  | 40 min |
> | L4GM                   | 0.1576       | -                   | 0.1932  | 25 sec |
> | V2M4                   | 0.1233       | 0.3023              | 0.2814  | 30 min |
> | GVFD                   | 0.3978       | -                   | 0.0699  | 10 min |
> | ShapeGen4D (Step1X-3D)            | 0.1220       | 0.3276              | 0.2934  | 3 min  |
> | ShapeGen4D (Hunyuan 2.1) | 0.0827 |  0.4155|  0.3971| 15 min|

---

> ### Author Response · Authors · 2025-11-22
>
> ### [W1] Generated results show flickering artifacts
> We agree with the reviewer that our current results exhibit some flickering artifacts. However, we believe our work represents an important step toward extending powerful 3D generators into the 4D domain. The progress in this field parallels the evolution from image generation to video generation over the past three years: early video models adapted from image generators also suffered from flickering, but with improved VAEs and large-scale training, smooth and realistic video generation became possible **without explicit physical representations or constraints**.
> We expect similar advancements for 4D generation. Our work demonstrates the feasibility of this approach, and we believe that with larger-scale training, better VAE designs, and modestly expanded datasets, we can achieve results for 4D generation analogous to those seen in video generation. In fact, we found that switching the base model from Step1X-3D (in the orginal draft) to a more powerful modeli, e.g., Hunyuan3D-2.1 noticeably reduced flickering artifacts. We have included visualizations of the updated ShapeGen4 results based on Hunyuan3D in **Figure 5**, as well as quantitative evaluations in **Table 1** (also shown in the table above), which demonstrate the improvements achieved by switching the base model.

---

> > ### Author Response · Authors · 2025-11-22
> >
> > ### [Q1] Incorporate data augmentation using geometric or spatial transformations to remove the requirement for global registration
> > We would like to clarify that the global registration is needed for aligning the pose of the generated shape with the input video. In **Section 4**, we explain that this requirement stems from the fact that the base pretrained 3D shape model was trained to generate 3D shapes agnostic to the viewpoint of the input image. Specifically, during training, an image from a random viewpoint is sampled, and the model learns to generate the 3D shape in a canonical (world) coordinate system, without applying any transformation relative to the input viewpoint. Due to this training scheme and the inherent ambiguity in defining the canonical pose, the generated 3D shape often exhibits a random orientation and is not guaranteed to align with the input view. This is why global registration is necessary, and **we don't see how applying additional image augmentations, on top of the already randomly sampled viewpoints, would help remove the need for global registration**.
> >
> > For completeness, we conduct the experiment suggested by the reviewer; Results in Table below. We did not observe significant improvement. To truly eliminate the need for global registration, large-scale retraining of the base model would be required, with 3D ground truth shapes explicitly transformed to match the input image viewpoints.
> >
> > | Method       | Chamfer↓ | IoU↑   | F-score↑ |
> > | ------------ | -------- | ------ | -------- |
> > | Baseline     | 0.1242   | 0.3249 | 0.2871   |
> > | Augmentation | 0.1255   | 0.3286 | 0.2815   |

---

### Author Response · Authors · 2025-11-22

We thank the reviewers for their constructive feedback and for recognizing the conceptual originality, simplicity, and computational efficiency of our approach. Below, we address each question and concern raised in detail.

---

### Author Response · Authors · 2025-12-03

We would like to once again thank the reviewers for their constructive and positive feedback. In our rebuttal, we believe we have addressed all questions and concerns raised. Notably:
1. We included additional visual examples demonstrating that our model can handle objects undergoing topology-changing animations.
2. We explained how our approach can be extended to long-sequence generation and presented a working procedure based on multi-diffusion, along with corresponding evaluations.
3. We clarified why we believe that learning motion implicitly—without relying on explicit deformation representations—is a promising direction, drawing an analogy to the evolution of video generation. We also highlighted that explicit motion representations can still be extracted in a post-processing step, as shown in the paper.
4. We added new results showing that applying our method on top of stronger 3D generation backbones yields improved quality and reduced temporal jitter.

---

### Meta-Review · Area_Chair_7Te4 · 2026-01-07

**Summary:**

Major weaknesses and issues raised by reviewers include

1) Technically lack of novelty

2) Lack of comparison like DreamMesh4D

3) To show examples with larger topology-changing animations

4) How to scale to deal with long videos

**Reviewer Concerns:**

All the comments as mentioned above should have been well addressed.

Note that although the long video experiment shows extra visual artefacts, this does not introduce killing issues while providing useful and insightful clues into the proposed model.

Comparing with the latest competitors in response further validates the performance advantage of this work.

**Reviewer Scores:**

Reviewer uhdb should be happy to raise the score to be positive as all the major issues have been well handled. Given that, this work would be rated to be positive overall. AC concurs that this work be of good value for the 4D generative AI community.

---

### Decision · Program_Chairs · 2026-01-26

Accept (Poster)